# APBench: A Unified Benchmark for Availability Poisoning Attacks and Defenses

## Abstract

The efficacy of availability poisoning, a method of poisoning data by injecting imperceptible perturbations to prevent its use in model training, has been a hot subject of investigation. Previous research suggested that it was difficult to effectively counteract such poisoning attacks. However, the introduction of various defense methods has challenged this notion. Due to the rapid progress in this field, the performance of different novel methods cannot be accurately validated due to variations in experimental setups. To further evaluate the attack and defense capabilities of these poisoning methods, we have developed a benchmark — APBench for assessing the efficacy of adversarial poisoning. APBench consists of 9 state-of-the-art availability poisoning attacks, 9 defense algorithms, and 4 conventional data augmentation techniques. We also have set up experiments with varying different poisoning ratios, and evaluated the attacks on multiple datasets and their transferability across model architectures. We further conducted a comprehensive evaluation of 2 additional attacks specifically targeting unsupervised models. Our results reveal the glaring inadequacy of existing attacks in safeguarding individual privacy. APBench is open source and available to the deep learning community[1].

## 1 Introduction

Recent advancements of deep neural networks (DNNs) [21, 37, 15] heavily rely on the abundant availability of data resources [5, 33, 19]. However, the unauthorized collection of large-scale data through web scraping for model training has raised concerns regarding data security and privacy. In response to these concerns, a new paradigm of practical and effective data protection methods has emerged, known as availability poisoning attacks (APA) [40, 45, 9, 17, 43, 10, 32, 14, 36, 8, 44, 14, 32], or unlearnable example attacks. These poisoning methods inject small perturbations into images that are typically imperceptible to humans, in order to hinder the model's ability to learn the original features of the images. Recently, the field of deep learning has witnessed advancements in defense strategies [23, 30, 7, 17] that hold the potential to challenge APAs, thereby undermining their claimed effectiveness and robustness. These defenses reveal the glaring inadequacy of existing APAs in safeguarding individual privacy in images. Consequently, we anticipate an impending arms race between attack and defense strategies in the near future.

However, evaluating the performance of these new methods across diverse model architectures and datasets poses a significant challenge due to variations in experimental settings of recent literatures. In addition, researchers face the daunting task of staying abreast of the latest methods and assessing the effectiveness of various competing attack-defense combinations. This could greatly hamper the development and empirical exploration of novel attack and defense strategies.

To tackle this challenge, we propose the APBench, a benchmark specifically designed for availability poisoning attacks and defenses. It involves implementing poisoning attack and defense mechanisms under standardized perturbations and training hyperparameters, in order to ensure fair and reproducible comparative evaluations. APBench comprises a range of availability poisoning attacks and defense algorithms, and commonly-used data augmentation policies. This comprehensive suite allows us to evaluate the effectiveness of the poisoning attacks thoroughly.

Our contributions can be summarized as follows:

---

[1]Link to follow

- An open source benchmark for state-of-the-art availability poisoning attacks and defenses, including 9 supervised and 2 unsupervised poisoning attack methods, 9 defense strategies and 4 common data augmentation methods.
- We conduct a comprehensive evaluation competing pairs of poisoning attacks and defenses.
- We conducted experiments across 4 publicly available datasets, and also extensively examined scenarios of partial poisoning, increased perturbations, the transferability of attacks to 4 CNN and 2 ViT models under various defenses, and unsupervised learning. We provide visual evaluation tools such as t-SNE, Shapley value map and Grad-CAM to qualitatively analyze the impact of poisoning attacks.

The aim of APBench is to serve as a catalyst for facilitating and promoting future advancements in both availability poisoning attack and defense methods. By providing a platform for evaluation and comparison, we aspire to pave the way for the development of future availability poisoning attacks that can effectively preserve utility and protect privacy.

## 2 RELATED WORK

### 2.1 AVAILABILITY POISONING ATTACKS

Availability poisoning attacks (APAs) belong to a category of data poisoning attacks [12] that adds a small perturbation to images, that is often imperceptible to humans. However, the objective contrasts with that of traditional data poisoning. The purpose of these perturbations is to **protect individual privacy** from deep learning algorithms, preventing DNNs from effectively learning the features present in the images. The attacker's goal is to thus render their data unlearnable with perturbations, hindering the unauthorized trainer from utilizing the data to learn models that can generalize effectively to the original data distribution. **The intent of APAs is therefore benign rather than malicious** as generally assumed of data poisoning attacks. We typically assume that the attacker publishes (a subset of) the images, which get curated and accurately labeled by the defender to train on them without consent from the attacker.

Formally, consider a source dataset comprising original examples $\mathcal{D}_{\text{clean}} = \{(\mathbf{x}_1, y_1), \ldots, (\mathbf{x}_n, y_n)\}$ where $\mathbf{x}_i \in \mathcal{X}$ denotes an input image and $y_i \in \mathcal{Y}$ represents its label. The objective of the attacker is thus to construct a set of availability perturbations $\boldsymbol{\delta}$, such that models trained on the set of *availability poisoned examples* $\mathcal{D}_{\text{poi}}(\boldsymbol{\delta}) = \{(\mathbf{x} + \boldsymbol{\delta}_{\mathbf{x}}, y) \mid (\mathbf{x}, y) \in \mathcal{D}_{\text{clean}}\}$ are expected to perform poorly when evaluated on a test set $\mathcal{D}_{\text{test}}$ sampled from the distribution $\mathcal{S}$:

$$\max_{\boldsymbol{\delta}} \mathbb{E}_{(\mathbf{x}_i, y_i) \sim \mathcal{D}_{\text{test}}} [\mathcal{L}(f_{\boldsymbol{\theta}^\star(\boldsymbol{\delta})}(\mathbf{x}_i), y_i)], \text{ s.t. } \boldsymbol{\theta}^\star(\boldsymbol{\delta}) = \operatorname*{argmin}_{\boldsymbol{\theta}} \mathbb{E}_{(\hat{\mathbf{x}}_i, y_i) \sim \mathcal{D}_{\text{poi}}(\boldsymbol{\delta})} \mathcal{L}(f_{\boldsymbol{\theta}}(\hat{\mathbf{x}}_i), y_i), \quad (1)$$

where $\mathcal{L}$ denotes the loss function, usually the softmax cross-entropy loss. In order to limit the impact on the original utility of images, the perturbation $\boldsymbol{\delta}_i$ is generally constrained within a small $\epsilon$-ball of $\ell_p$ distance.

To enforce a small perturbation budget, recent methods typically constrain their perturbations within a small $\ell_p$-ball of $\epsilon$ radius, where typically $p \in \{0, 2, \infty\}$. DeepConfuse (DC) [8] proposes to use autoencoders to generate training-phase adversarial perturbations. Neural tangent generalization attacks (NTGA) [45] approximates the target model as a Gaussian process [18] using the generalized neural tangent kernel, and solves a bi-level optimization for perturbations. Error-minimizing attacks (EM) [17] minimizes the training error of the perturbed images relative to their original labels on the target model, creating shortcuts for the data to become "unlearnable" by the target model. Building upon EM, robust error-minimizing attacks (REM) [10] use adversarially trained models to generate perturbations in order to counter defense with adversarial training. Hypocritical [40] also generates error-minimizing perturbations similar to EM, but instead uses a pretrained surrogate model. Targeted adversarial poisoning (TAP) [9], inspired by [26], found adversarial examples could be used for availability poisoning. In contrast to the above approaches, indiscriminate poisoning (UCL) [14] and transferable unlearnable examples (TUE) [32] instead consider availability poisoning for unsupervised learning. On the other hand, $\ell_2$ and $\ell_0$ perturbation-based poisoning methods do not require a surrogate model. They achieve poisoning by searching for certain triggering patterns to create shortcuts in the network. Besides the above $\ell_\infty$-bounded methods, Linear-separable poisoning (LSP) [44] and Autoregressive Poisoning (AR) [36] both prescribe perturbations within

an $\ell_2$ perturbation budget. Specifically, LSP generates randomly initialized linearly separable color block perturbations, while AR fills the starting rows and columns of each channel with Gaussian noise and uses an autoregressive process to fill the remaining pixels, generating random noise perturbations. One Pixel Shortcut [43] (OPS), as an $\ell_0$-bounded poisoning method, perturbs only a single pixel in the training image to achieve strong poisoning in terms of usability. Figure 1 provides visual examples of these attacks.

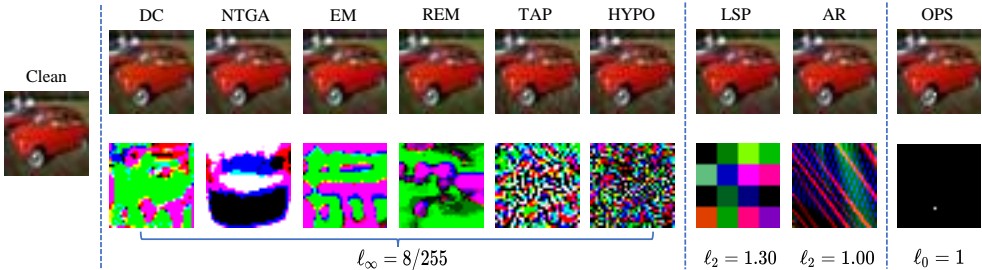

Figure 1: Visualizations of unlearnable CIFAR-10 images with corresponding perturbations. Perturbations are normalized for visualization.

## 2.2 AVAILABILITY POISONING DEFENSES

The goal of the defender is to successfully train a model with good generalization abilities (*e.g.*, test accuracies on natural unseen images) on protected data. Generally, the defender can control the training algorithm, and only have access to a training data set with data poisoned either partially or fully. The objective of the defender is thus to find a novel training algorithm $g(\mathcal{D}_{\text{poi}})$ that trains models to generalize well to the original data distribution:

$$\min_g \mathbb{E}_{(\mathbf{x}_i, y_i) \sim \mathcal{D}_{\text{test}}}[\mathcal{L}(f_{\boldsymbol{\theta}^\star}(\mathbf{x}_i), y_i)], \text{ s.t. } \boldsymbol{\theta}^\star = g(\mathcal{D}_{\text{poi}}). \tag{2}$$

Notably, if the method employs the standard training loss but performs novel image transformations $h$, then $g$ can be further specialized as follows:

$$g(\mathcal{D}_{\text{poi}}) = \operatorname{argmin}_{\boldsymbol{\theta}} \mathbb{E}_{(\hat{\mathbf{x}}_i, y_i) \sim \mathcal{D}_{\text{poi}}(\boldsymbol{\delta})} \mathcal{L}(f_{\boldsymbol{\theta}}(h(\hat{\mathbf{x}}_i)), y_i). \tag{3}$$

Currently, defense methods against perturbative availability poisoning can be mainly classified into two categories: preprocessing and training-phase defenses. Data preprocessing methods preprocess the training images to eliminate the poisoning perturbations prior to training. Image shortcuts squeezing (ISS) [23] consists of simple countermeasures based on image compression, including grayscale transformation, JPEG compression, or bit-depth reduction (BDR) to perform poison removal. Recently, AVATAR [7] leverages the method proposed in DiffPure [28] to employ diffusion models to disrupt deliberate perturbations while preserving semantics in the training images. On the other hand, training-phase defense algorithms apply specific modifications to the training phase to defense against availability attacks. Adversarial training has long been considered the most effective defense mechanism [17, 10] against such attacks. Recent report [35] finds that peak accuracy can be reached early in the training of availability poisons, and thus early stopping can be an effective mean of training-phase defense. Adversarial augmentations [30] sample multiple augmentations on one image, and train models on the maximum loss of all augmented images to prevent learning from poisoning shortcuts. For referential baselines, APBench also includes commonly used data augmentation techniques such as Gaussian blur, random crop and flip (standard training), CutOut [6], CutMix [46], and MixUp [47], and show their (limited) effect in mitigating availability poisons.

## 2.3 RELATED BENCHMARKS

Availability poisoning is closely connected to the domains of *adversarial* and *backdoor* attack and defense algorithms. Adversarial attacks primarily aim to deceive models with adversarial perturbations during inference to induce misclassifications. There are several libraries and benchmarks available for evaluating adversarial attack and defense techniques, such as Foolbox [31], AdvBox [13], and RobustBench [4].

*Backdoor* or *data poisoning* [3] attacks focus on injecting backdoor triggers into the training algorithm or data respectively, causing trained models to misclassify images containing these triggers while maintaining or minimally impacting clean accuracy. In contrast to APAs, such attacks introduce hidden behaviors into the model that can be triggered by specific inputs, often for malicious purposes. Benchmark libraries specifically designed for backdoor attacks and defenses include TrojanZoo [29], Backdoorbench [42], and Backdoorbox [22]. Moreover, [38, 11] introduce benchmarks and frameworks for data poisoning attacks.

However, there is currently a lack and an urgent need of a dedicated and comprehensive benchmark that standardizes and evaluates availability poisoning attack and defense strategies. To the best of our knowledge, APBench is the first benchmark that fulfills this purpose. It offers an extensive library of recent attacks and defenses, explores various perspectives, including the impact of poisoning rates and model architectures, as well as attack transferability. We hope that APBench can make significant contributions to the community and foster the development of future availability attacks for effective privacy protection.

## 3 A UNIFIED AVAILABILITY POISONING BENCHMARK

As shown in Figure 2, APBench consists of three main components: (a) The availability poisoning attack module. This library includes a set of representative availability poisoning attacks that can generate unlearnable versions of a given clean dataset. (b) The poisoning defense module. This module integrates a suite of state-of-the-art defenses that can effectively mitigate the unlearning effect and restore clean accuracies to a certain extent. (c) The evaluation module. This module can efficiently analyze the performance of various availability poisoning attack methods using accuracy metrics and visual analysis strategies.

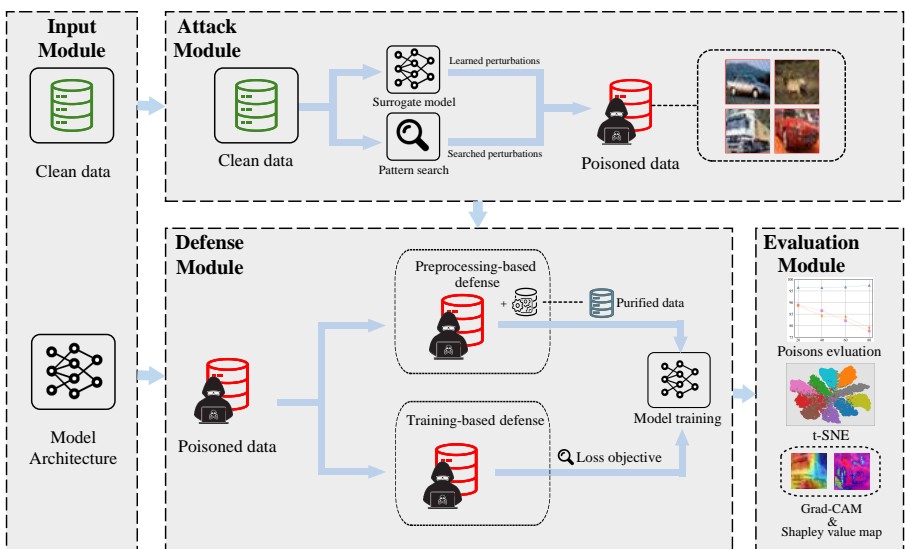

Figure 2: The overall system design of APBench.

We built an extensible codebase as the foundation of APBench. In the attack module, we provide a total of 9 availability poisoning attacks of 3 different perturbation types ($\ell_p$) for supervised learning, and 2 attacks for unsupervised learning. For each availability poisoning attack method, we can generate their respective poisoned datasets. This module also allows us to further expand to different perturbations budgets, poisoning ratios, and easily extend to future poisoning methods. Using the poisoned datasets generated by the attack module, we can evaluate defenses through the defense module. The goal of this module is to ensure that models trained on unlearnable datasets can still generalize well on clean data. The defense module primarily achieves poisoning mitigation through data preprocessing or training-phase defenses. Finally, the evaluation module computes the accuracy

Table 1: Availability poisoning attack algorithms implemented in APBench. "Type" and "Budget" respectively denotes the type of perturbation and its budget. "Mode" denotes the training mode, where "S" and "U" and respectively mean supervised and unsupervised training. "No surrogate" denotes whether the attack requires access to a surrogate model for perturbation generation. "Class-wise" and "Sample-wise" indicate if the attack supports class-wise and sample-wise perturbation generation. "Stealthy" denotes whether the attack is stealthy to human.

| Attack Method | Type | Budget | Mode | No surrogate | Class-wise | Sample-wise | Stealthy |
|---|---|---|---|---|---|---|---|
| DC [8] | | | S | | | ✓ | ✓ |
| NTGA [45] | | | S | | | ✓ | ✓ |
| HYPO [40] | | | S | | | ✓ | ✓ |
| EM [17] | $\ell_\infty$ | 8/255 | S | | ✓ | ✓ | ✓ |
| REM [10] | | | S | | ✓ | ✓ | ✓ |
| TAP [9] | | | S | | | ✓ | ✓ |
| UCL [14] | | | U | | ✓ | ✓ | ✓ |
| TUE [32] | | | U | | | ✓ | ✓ |
| LSP [44] | $\ell_2$ | 1.30 | S | ✓ | ⊙ | ✓ | |
| AR [36] | $\ell_2$ | 1.00 | S | ✓ | ⊙ | ✓ | ✓ |
| OPS [43] | $\ell_0$ | 1 | S | ✓ | ✓ | | |

Table 2: Availability poisoning defense algorithms implemented in APBench.

| Defense Method | Type | Time Cost | Description |
|---|---|---|---|
| Standard | | Low | Random image cropping and flipping |
| CutOut [6] | Data augmentations | Low | Random image erasing |
| MixUp [47] | | Low | Random image blending |
| CutMix [46] | | Low | Random image cutting and stitching |
| Gaussian (used in [23]) | | Low | Image blurring with a Gaussian kernel |
| BDR (used in [23]) | | Low | Image bit-depth reduction |
| Gray (used in [23]) | Data preprocessing | Low | Image grayscale transformation |
| JPEG (used in [23]) | | Low | Image compression |
| AVATAR [7] | | High | Image corruption and restoration |
| Early stopping [35] | | Low | Finding peak validation accuracy |
| UEraser-Lite [30] | Training-phase defense | Low | Stronger data augmentations |
| UEraser-Max [30] | | High | Adversarial augmentations |
| AT [25] | | High | Adversarial training |

metrics of different attacks and defense combinations, and can also perform qualitative visual analyses to help understand the characteristics of the datasets.

Our benchmark currently includes 9 supervised and 2 unsupervised availability poisoning attacks, 9 defense algorithms, and 4 traditional image augmentation methods. In Table 1 and Table 2, we provide a brief summary of the properties of attack and defense algorithms. More detailed descriptions for each algorithm are provided in Appendix B.

## 4 EVALUATIONS

**Datasets** We evaluated our benchmark on 4 commonly used datasets (CIFAR-10 [20], CIFAR-100 [20], SVHN [27], and an ImageNet [5] subset) and 5 mainstream models (ResNet-18 [15], ResNet-50 [15], MobileNetV2 [34], and DenseNet-121 [16]). To ensure a fair comparison between attack and defense methods, we used only the basic version of training for each model. Appendix A summarizes the specifications of the datasets and the test accuracies achievable through standard training on clean training data, and further describes the detail specifications of each dataset.

**Attacks and defenses** We evaluated combinations of availability poisoning attacks and defense methods introduced in Section 3. Moreover, we explored 5 different data poisoning rates and 5 different models. In addition, We also explore two availability poisonings for unsupervised learning (UCL [14] and TUE [32]) and evaluate them on the recently proposed defenses (Gray, JPEG, Early

stopping (ES), UEraser-Lite [30], and AVATAR [7]). The implementation details of all algorithms and additional results can be found in Appendix B.

**Types of Threat Models** We can classify adversarial attacks based on three distinct availability poisoning threat models: $\ell_\infty$-bounded attacks (DC, NTGA, EM, REM, TAP, and HYPO); $\ell_2$-bounded attacks (LSP and AR); an $\ell_0$-bounded attack (OPS). Given that $\ell_0$ perturbations resist disruption from image preprocessing or augmentations and remain unaffected by $\ell_\infty$ adversarial training, the $\ell_0$-bounded OPS attack demonstrates robustness against a plethora of defenses. Conversely, in terms of stealthiness, the $\ell_0$ attacks are less subtle than their $\ell_\infty$ and $\ell_2$ counterparts, as illustrated in Figure 1. Perturbations bounded by both $\ell_\infty$ and $\ell_2$ are comparable *w.r.t.* the degree of visual stealthiness and effectiveness. Importantly, the two $\ell_2$-bounded attacks (LSP and AR) do not require surrogate model training, and are thus more efficient in the unlearnable examples synthesis.

**Training settings** We trained the CIAFR-10, CIFAR-100 and ImageNet-subset models for 200 epochs and the SVHN models for 100 epochs. We used the stochastic gradient descent (SGD) optimizer with a momentum of 0.9 and a learning rate of 0.1 by default. As for unsupervised learning, all experiments are trained for 500 epochs with the SGD optimizer. The learning rate is 0.5 for SimCLR [1] and 0.3 for MoCo-v2 [2]. Please note that we generate sample-wise perturbations for all availability poisoning attacks. Specific settings for each defense method may have slight differences, and detailed information can be found in the Appendix C.

**Standard Scenario** To start, we consider a common scenario where both the surrogate model and target model are ResNet-18, and the poisoning rate is set to 100%. We first evaluate the performance of the supervised poisoning methods against 4 state-of-the-art defense mechanisms and 4 commonly used data augmentation strategies. Table 3 presents the evaluation results on CIFAR-10 from our benchmark. It is evident that the conventional data augmentation methods appear to be ineffective against all poisoning methods. Yet, *even simple image compression methods (BDR, grayscale, and JPEG corruption from ISS [23]) demonstrate a notable effect in mitigating the poisoning attacks*, but fails to achieve high clean accuracy. Despite requiring more computational cost or additional resources (pretrained diffusion models for AVATAR), methods such as UEraser-Max [30] and AVATAR [7], generally surpass the image compression methods from ISS in terms of effectiveness. While AVATAR is inferior to UEraser-Max in gaining accuracy, it decouples the defense into an independent data sanitization phase, allowing it to be directly used in all existing training scenarios. While the early stopping (ES) method can be somewhat effective as a defense, is not usually considered a good one. mainly due to the fact that the peak accuracy of the availability poisoning is not ideal. Adversarial training appears effective but in many cases is outperformed by even a simple JPEG compression, it also fails notably against OPS, as the $\ell_\infty$ perturbation budget cannot mitigate $\ell_0$ threats. We further conduct experiments on the CIFAR-100, SVHN, and ImageNet-subset datasets, and the results are shown in Table 4.

Our findings indicate that perturbations constrained by traditional $\ell_p$ norms are ineffective against adversarial augmentation (UEraser-Max), and image restoration by pretrained diffusion models (AVATAR), as they break free from the assumption of $\ell_p$ constraints. Even simple image compression techniques (JPEG, Grayscale, and BDR) can effectively remove the effect of perturbations. At this stage, availability poisoning attacks that rely on $\ell_p$-bounded perturbations may not be as effective as initially suggested by the relevant attacks.

## 4.1 CHALLENGING SCENARIOS

To further investigate the effectiveness and robustness of availability poisoning attacks and defenses, we conducted evaluations in more challenging scenarios. We considered partial poisoning scenarios, larger perturbation poisoning, and the attack transferability to different models.

**Partial poisoning** In realistic scenarios, it is difficult for an attacker to achieve modification of the entire dataset. We thus investigate the impact of poisoning rate on the performance of availability poisoning. Figure 3 presents the results on CIFAR-10 and ResNet-18, *w.r.t.* each poisoning rate for attack-defense pairs, where each subplot corresponds to a specific poisoning attack method. We explore four different poisoning rates (20%, 40%, 60%, 80%).

**Privacy protection under partial poisoning** As can be seen in Figure 3, the test accuracy of the model in the case of partial poisoning is only slightly lower than that in the case of a completely

Table 3: Test accuracies (%) of models trained on poisoned CIFAR-10 datasets. The model trained on a clean CIFAR-10 dataset attains an accuracy of 94.32%.

| Method | Standard | CutOut | CutMix | MixUp | Gaussian | BDR | Gray | JPEG | ES | U-Max | AVATAR | AT |
|---|---|---|---|---|---|---|---|---|---|---|---|---|
| DC | 15.19 | 19.94 | 17.91 | 25.07 | 16.10 | 67.73 | 85.55 | 83.57 | 26.08 | 92.17 | 82.10 | 76.85 |
| EM | 20.78 | 18.79 | 22.28 | 31.14 | 14.71 | 37.94 | 92.03 | 80.72 | 25.39 | 93.61 | 75.62 | 82.51 |
| REM | 17.47 | 21.96 | 26.22 | 43.07 | 21.80 | 58.60 | 92.27 | 85.44 | 31.32 | 92.43 | 82.42 | 77.46 |
| HYPO | 70.38 | 69.04 | 67.12 | 74.25 | 62.17 | 74.82 | 63.35 | 85.21 | 70.52 | 88.44 | 85.94 | 81.49 |
| NTGA | 22.76 | 13.78 | 12.91 | 20.59 | 19.95 | 59.32 | 70.41 | 68.72 | 28.19 | 86.78 | 86.22 | 69.70 |
| TAP | 6.27 | 9.88 | 14.21 | 15.46 | 7.88 | 70.75 | 11.01 | 84.08 | 39.54 | 79.05 | 87.75 | 79.92 |
| LSP | 13.06 | 14.96 | 17.69 | 18.77 | 18.61 | 53.86 | 64.70 | 80.14 | 29.10 | 92.83 | 76.90 | 81.38 |
| AR | 11.74 | 10.95 | 12.60 | 14.15 | 13.83 | 36.14 | 35.17 | 84.75 | 44.29 | 90.12 | 88.60 | 81.15 |
| OPS | 14.69 | 52.98 | 64.72 | 49.27 | 13.38 | 37.32 | 19.88 | 78.48 | 38.20 | 77.99 | 66.16 | 14.95 |

Table 4: Test accuracies (%) on poisoned CIFAR-100, SVHN and ImageNet-subset datasets.

| Dataset | Method | Standard | CutOut | CutMix | MixUp | Gaussian | BDR | Gray | JPEG | ES | U-Max |
|---|---|---|---|---|---|---|---|---|---|---|---|
| CIFAR-100 | EM | 3.03 | 4.15 | 3.98 | 6.46 | 2.99 | 34.10 | 59.14 | 58.71 | 7.06 | 68.81 |
| | REM | 3.73 | 4.00 | 3.71 | 10.90 | 3.59 | 29.16 | 57.47 | 55.60 | 10.99 | 67.72 |
| | LSP | 2.56 | 2.33 | 4.52 | 4.86 | 1.71 | 27.12 | 39.45 | 52.82 | 9.52 | 68.31 |
| | AR | 1.87 | 1.63 | 3.17 | 2.35 | 2.62 | 31.15 | 16.13 | 54.73 | 26.58 | 55.95 |
| SVHN | EM | 10.33 | 13.38 | 10.77 | 12.79 | 8.82 | 36.65 | 65.66 | 86.14 | 13.47 | 90.24 |
| | REM | 14.02 | 18.92 | 9.55 | 19.56 | 7.54 | 42.52 | 19.59 | 90.58 | 19.61 | 88.26 |
| | LSP | 12.16 | 12.98 | 8.17 | 18.86 | 7.15 | 26.67 | 16.90 | 84.06 | 12.91 | 90.64 |
| | AR | 19.23 | 14.92 | 6.71 | 13.52 | 7.75 | 39.24 | 10.00 | 92.46 | 89.32 | 90.07 |
| ImageNet-100 | EM | 2.94 | 4.05 | 4.73 | 4.15 | 3.15 | 6.45 | 12.20 | 31.73 | 8.80 | 44.07 |
| | REM | 3.66 | 4.13 | 4.78 | 3.94 | 4.28 | 4.03 | 3.95 | 40.98 | 17.19 | 42.14 |
| | LSP | 38.52 | 40.56 | 29.78 | 7.85 | 42.68 | 26.58 | 25.18 | 36.83 | 39.52 | 63.28 |

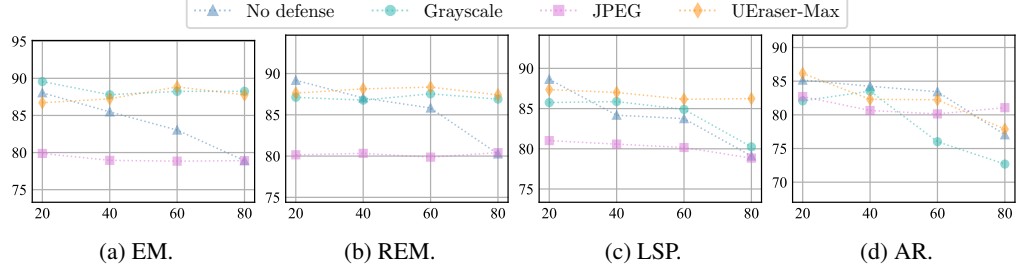

(a) EM.                (b) REM.                (c) LSP.                (d) AR.

Figure 3: The efficacy in test accuracies (%, vertical axes) of defenses (No defense, Grayscale, JPEG, and UEraser-Max) against different partial poisoning attacks including EM (a), REM (b), LSP (c), and AR (d) with poisoning ratios (horizontal axes) ranging from 20% to 80%.

clean dataset. This raises the following question: *Are APAs effective in protecting only a portion of the training data?* To answer, we introduce poisoning perturbations with APAs to a varying portion of the training data, and investigate how well the models learn the origin features that exist in the poisoned images for different poisoning rates. For this, Figure 4 evaluates and compares the mean losses of the unlearnable images used during training ("Unlearnable"), the origin images of the unlearnable part ("Clean"), and for reference, the mean losses of images unseen by the model from the test set ("Test"), and "Train" means the loss of the clean part of the training set. We find that the losses on the original images of the unlearnable part is similar to that of the test set, or even lower. This suggests that **the availability poisoning perturbations can reasonably protect the private data against undefended learning**. For a similar comparison of accuracies, please refer to Appendix C.1.

**Larger perturbations** We increased the magnitude of perturbations in availability poisoning attacks to further evaluate the performance of attacks and defenses. Table 5 presents the results of availability poisoning with larger perturbations on CIFAR-10. Due to such significant perturbations, their stealthiness is further reduced, making it challenging to carry out such attacks in realistic scenarios. However, larger perturbations indeed have a more pronounced impact on suppressing defense performance, leading to significant accuracy losses for all defense methods. There exists a trade-off between perturbation magnitude and accuracy recovery. Considering that at larger perturbations, availability poisoning is dramatically less stealthy, and some defense methods are still effective, it is not recommended to use larger perturbations.

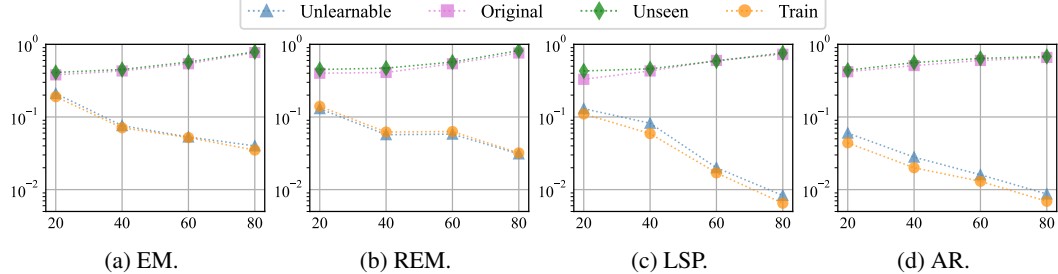

(a) EM. (b) REM. (c) LSP. (d) AR.

Figure 4: The mean losses (vertical axes) indicate that original features in unlearnable examples are not learned by the model. All evaluations consider partial poisoning scenarios (poisoning rates from 20% to 80%, horizontal axes). Note that "Unlearnable" and "Original" respectively denote the set of unlearnable examples, and their original clean variants, "Train" means the loss of the clean part of the training set. and "Unseen" denote images from the test set unobserved during model training.

Table 5: Test accuracies (%) on poisoned CIFAR-10 datasets with increased perturbations.

| Method | Budget | No defense | Gray | JPEG | ES | U-Max | AT |
|---|---|---|---|---|---|---|---|
| EM | $\ell_\infty = 16/255$ | 18.74 | 76.76 | 55.96 | 27.39 | 88.09 | 77.82 |
| REM | $\ell_\infty = 16/255$ | 19.80 | 83.65 | 80.07 | 33.07 | 80.36 | 75.64 |
| LSP | $\ell_2 = 1.74$ | 15.83 | 37.60 | 42.83 | 27.30 | 87.20 | 77.92 |
| AR | $\ell_2 = 1.50$ | 11.20 | 26.10 | 78.24 | 20.96 | 68.42 | 70.14 |

Table 6: Clean test accuracies of different CIFAR-10 target models, where attacks are oblivious to the model architectures. Note that AR and LSP are surrogate-free, and for EM and REM the surrogate model is ResNet-18.

| Model | Clean | Method | No defense | Gray | JPEG | ES | U-Max | AVATAR |
|---|---|---|---|---|---|---|---|---|
| ResNet-50 | 94.47 | EM | 14.41 | 83.40 | 76.88 | 26.69 | 85.89 | 77.64 |
| | | REM | 16.26 | 87.26 | 75.79 | 31.37 | 92.69 | 83.68 |
| | | LSP | 19.23 | 68.94 | 73.24 | 32.73 | 93.08 | 76.47 |
| | | AR | 11.83 | 27.51 | 80.24 | 28.66 | 81.40 | 86.39 |
| SENet-18 | 94.83 | EM | 13.60 | 86.03 | 79.35 | 16.35 | 83.27 | 74.22 |
| | | REM | 20.99 | 84.50 | 78.92 | 22.85 | 93.17 | 84.37 |
| | | LSP | 18.54 | 65.06 | 76.51 | 26.38 | 92.53 | 75.19 |
| | | AR | 13.68 | 34.26 | 79.29 | 37.04 | 75.06 | 84.37 |
| MobileNetV2 | 94.62 | EM | 15.62 | 77.21 | 70.96 | 16.71 | 82.71 | 75.62 |
| | | REM | 20.83 | 80.81 | 72.27 | 21.92 | 91.03 | 82.77 |
| | | LSP | 16.82 | 61.07 | 72.03 | 28.12 | 92.10 | 76.81 |
| | | AR | 13.36 | 28.54 | 68.14 | 39.45 | 73.40 | 81.63 |
| DenseNet-121 | 95.08 | EM | 13.89 | 82.49 | 78.42 | 15.68 | 82.37 | 76.69 |
| | | REM | 21.45 | 85.47 | 78.42 | 22.35 | 93.09 | 83.04 |
| | | LSP | 18.94 | 67.95 | 74.90 | 26.86 | 93.47 | 78.22 |
| | | AR | 13.43 | 25.51 | 81.12 | 36.51 | 82.36 | 89.92 |
| ViT-small | 84.66 | EM | 21.47 | 80.42 | 72.64 | 30.91 | 74.29 | 54.84 |
| | | REM | 32.17 | 79.65 | 74.92 | 43.07 | 83.27 | 73.57 |
| | | LSP | 29.06 | 59.34 | 68.07 | 32.69 | 87.01 | 66.74 |
| | | AR | 25.04 | 38.90 | 74.77 | 45.54 | 63.90 | 78.64 |
| CaiT-small | 71.96 | EM | 17.01 | 64.76 | 63.75 | 39.69 | 63.37 | 41.94 |
| | | REM | 26.11 | 65.05 | 66.43 | 47.39 | 72.05 | 62.53 |
| | | LSP | 25.08 | 63.06 | 57.15 | 37.95 | 70.92 | 51.39 |
| | | AR | 68.63 | 66.27 | 69.30 | 67.41 | 70.04 | 62.77 |

**Attack transferability across models** In real-world scenarios, availability poisoning attackers can only manipulate the data and do not have access to specific details of the defender. Therefore, we conducted experiments on different model architectures. It is worth noting that all surrogate-based attack methods are considered using ResNet-18. The results are shown in Table 6. It is evident that all surrogate-based and -free poisoning methods exhibit strong transferability, while the three recently proposed defenses also achieve successful defense across different model architectures. The only exception is the AR method, which fails against CaiT-small.

Table 7: Test accuracies (%) of adaptive poisoning with EM on ResNet-18.

| Method | Standard | Gray | JPEG | U-Max |
|---|---|---|---|---|
| EM + Gray | 19.48 | 21.64 | 78.39 | 90.52 |
| EM + JPEG | 20.67 | 90.29 | 76.25 | 93.22 |
| EM + UEraser | 35.24 | 88.62 | 80.46 | 89.55 |

Table 8: Test accuracies (%) of adaptive poisoning with REM on ResNet-18.

| Method | Standard | Gray | JPEG | U-Max |
|---|---|---|---|---|
| REM + Gray | 16.70 | 56.33 | 82.47 | 91.37 |
| REM + JPEG | 19.45 | 91.71 | 75.84 | 92.53 |
| REM + UEraser | 21.61 | 89.26 | 77.51 | 91.84 |

**Adaptive poisoning** We evaluated strong adaptive poisons against various defenses using two poisoning methods, EM [17] and REM [10]. We assume that the defenders can be adapted to three defenses (Gray, JPEG, and UEraser), by using the attack in the perturbation generation process. From Tables 7 and 8, it can be seen that adaptive poisoning significantly affects the performance of the Gray defense, but has less effect on JPEG and UEraser.

**Unsupervised learning** We evaluated the availability poisoning attacks targeting unsupervised models on CIFAR-10. We considered two popular unsupervised learning frameworks: SimCLR [1] and MoCo-v2 [2]. All defense methods were applied before the data augmentation process, which means they were applied to preprocessed images before undergoing different data augmentations. Therefore, we only applied UEraser-Lite as a data preprocessing method. The results of all experiments are shown in Table 9.

Table 9: Performance of availability poisoning attacks and defense on different unsupervised learning algorithms and datasets. Note that "U-Lite" denotes UEraser-Lite.

| Algorithm | Method | No Defense | Gray | JPEG | U-Lite | AVATAR |
|---|---|---|---|---|---|---|
| SimCLR | UCL | 47.25 | 46.91 | 66.76 | 68.42 | 83.22 |
|  | TUE | 57.10 | 56.37 | 67.54 | 66.59 | 84.24 |
| MoCo-v2 | UCL | 53.78 | 53.34 | 65.44 | 72.13 | 83.08 |
|  | TUE | 66.73 | 64.95 | 67.28 | 74.82 | 82.48 |

**Visual analyses** We provide visualization tools (Grad-CAM [39] and Shapley value maps [24]) to facilitate the analysis and understanding of availability poisoning attacks. We also use t-SNE [41] to visualize the availability poisons (Figure 7). Although t-SNE cannot accurately represent high-dimensional spaces, it aids in the global visualization of feature representations, allowing us to observe specific characteristics of availability poisons. For additional discussions on the visualizations, please refer to Appendix C.3.

**Future outlook** Future research directions on APAs should explore methods that enhance the resilience of perturbations. One approach to consider is the development of **generalizable attacks**, which can simultaneously target the DNNs being trained, diffusion models for image restoration, and remain robust against traditional or **color distortions**, among others. On the other hand, semantic-based perturbations offer an alternative strategy, as such modifications to images can be challenging to remove by defenses.

## 5 CONCLUSIONS

We have established the first comprehensive and up-to-date benchmark for the field of availability poisoning, covering a diverse range of availability poisoning attacks and state-of-the-art defense algorithms. We have conducted effective evaluations and analyses of different combinations of attacks and defenses, as well as additional challenging scenarios. Through this new benchmark, our primary objective is to provide researchers with a clearer understanding of the current progress in the field of availability poisoning attacks and defenses. We hope it can enable rapid comparisons between existing methods and new approaches, while also inspiring fresh ideas through our comprehensive benchmark and analysis tools. We believe that our benchmark will contribute to the advancement of availability poisoning research and the development of more effective methods to safeguard privacy.

## 6 REPRODUCIBILITY STATEMENT

We provide an open-source implementation of all attacks and defenses in the supplementary material. Following the README file, users can run all experiments on their own device to reproduce the results shown in paper.

## 7 ETHICS STATEMENT

Similar to many other technologies, the implementation of availability poisoning algorithms can be used by users for both beneficial and malicious purposes. We understand that these poisoning attack methods were originally proposed to protect privacy, but they can also be used to generate maliciously data to introduce model backdoors. The benchmark aims to promote an understanding of various availability poisoning attacks and defense methods, as well as encourage the development of new algorithms in this field. It is also important for us to raise awareness of the false sense of security provided by availability poisoning attacks. However, we emphasize that the use of these algorithms and evaluation results should comply with ethical guidelines and legal regulations. We encourage users to be aware of the potential risks of the technology and take appropriate measures to ensure its beneficial use for both society and individuals.

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

# A  DATASETS

Table 10 summarizes the specifications of datasets and the respective test accuracies of typical training on ResNet-18 architectures.

Table 10: Dataset specifications and the respective test accuracies on ResNet-18.

| Datasets | #Classes | Training / Test Size | Image Dimensions | Clean Accuracy (%) |
|----------|----------|----------------------|------------------|--------------------|
| CIFAR-10 [20] | 10 | 50,000 / 10,000 | $32\times32\times3$ | 94.32 |
| CIFAR-100 [20] | 100 | 50,000 / 10,000 | $32\times32\times3$ | 75.36 |
| SVHN [27] | 10 | 73,257 / 26,032 | $32\times32\times3$ | 96.03 |
| ImageNet-subset [5] | 100 | 20,000 / 4,000 | $224\times224\times3$ | 64.18 |

# B  IMPLEMENTATION DETAILS

In addition to the discussion of properties of the availability poisoning attacks and defenses presented in Tables 1 and 2, here, we provide a high-level description of the attack and defense algorithms implemented in APBench.

**Attacks**:

- **Deep Confuse (DC)** [8]: DC is proposed as a novel approach to manipulating classifiers by modifying the training data. Its key idea involves employing an autoencoder-like network to capture the training trajectory of the target model and adversarially perturbing the training data.

- **Error-minimizing attack (EM)** [17]: EM trains a surrogate model by minimizing the error of images relative to their original labels, generating perturbations that minimize the errors and thus render the perturbed images unlearnable. The authors of EM introduce the threat model of availability poisoning attacks, highlighting their role as a mechanism for privacy protection.

- **Neural tangent generalization attack (NTGA)** [45]: NTGA simulates the training dynamics of a generalized deep neural network using a Gaussian process and leverages this surrogate to find better local optima with improved transferability.

- **Hypocritical (HYPO)** [40]: HYPO, similar to EM, generates images that minimize errors relative to their true labels using a pre-trained model.

- **Targeted adversarial poisoning (TAP)** [9]: TAP achieves availability poisoning by generating targeted adversarial examples of non-ground-truth labels of pre-trained models.

- **Robust error-minimizing attacks (REM)** [10]: REM improves the poisoning effect of availability poisoning by replacing the training process of the surrogate model with adversarial training.

- **Linear-separable poisoning (LSP)** [44]: LSP generates randomly initialized linearly separable color block perturbations, enabling effective availability poisoning attacks without requiring surrogate models or excessive computational overhead.

- **Autoregressive Poisoning (AR)** [36]: AR, similar to LSP, does not require additional surrogate models. It fills the initial rows and columns of each channel with Gaussian noise and uses an autoregressive process to fill the remaining pixels, generating random noise perturbations.

- **One-Pixel-Shortcut (OPS)** [43]: OPS is a targeted availability poisoning attack that perturbs only one pixel of an image, generating an effective availability poisoning attack against traditional adversarial training methods.

- **Indiscriminate poisoning (UCL)** [14]: UCL considers generating unlearnable examples for unsupervised learning by minimizing the CL loss (*e.g.*, the InfoNCE loss) in the unsupervised learning setting.

- **Transferable unlearnable examples (TUE)** [32]: TUE discovers that UCL is effective only in unsupervised learning, while its performance significantly deteriorates in supervised learning. Therefore, TUE is proposed that simultaneously targets both supervised and unsupervised learning. Different to UCL, it additionally embeds linear separable poisons into unsupervised unlearnable examples using the class-wise separability discriminant.

**Defenses**:

- **Adversarial training (AT)** [25]: AT is a widely-recognized effective approach against availability poisoning. Small adversarial perturbations are applied to the training images during training, in order to improve the robustness of the model against perturbations.
- **Image Shortcut Squeezing (ISS)** [23]: ISS uses traditional image compression techniques such as grayscale transformation, bit-depth reduction (BDR), and JPEG compression, as defenses against availability poisoning.
- **Early stopping (ES)** [35]: Early stopping can quickly achieve peak accuracy on availability poisons, but due to the difference in behavior of various poisons, it fails to achieve favorable defense results.
- **Adversarial augmentations (UEraser)** [30]: UEraser-Lite uses an effective augmentation pipeline to suppress availability poisoning shortcuts. UEraser-Max further improves the defense against availability poisoning through adversarial augmentations.
- **AVATAR** [7]: Following DiffPure [28], AVATAR cleans the images of the unlearnable perturbations with diffusion models.

## C EXPERIMENTAL SETTINGS AND ADDITIONAL RESULTS

Table 11 presents the default hyperparameters for all availability poisoning attacks implemented in APBench.

### C.1 PARTIAL POISONING

In addition to the discussion on partial poisoning in Section 4, we provide the results in terms of accuracies in Figure 5.

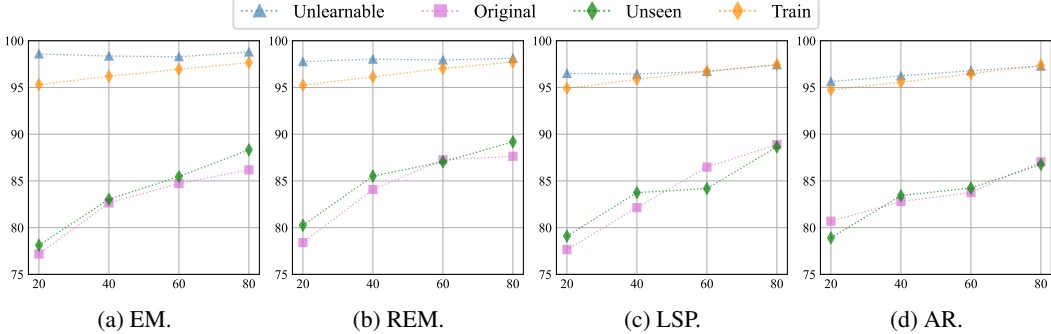

Figure 5: The accuracies (%, vertical axes) indicate that original features in unlearnable examples are not learned by the model. All evaluations consider partial poisoning scenarios (poisoning rates from 20% to 80%, horizontal axes). Note that "Unlearnable" and "Original" respectively denote the set of unlearnable examples, and their original clean variants, and "Unseen" denote images from the test set unobserved during model training.

### C.2 TEST ROBUSTNESS

Table 14 additionally compared the test accuracy of the PGD-20 adversarial examples on CIFAR-10 with a perturbation budget of $8/255$. Note that all experiments considered both the white-box scenario, and black-box transferability from a different initialization, and the step size is set to $2/255$.

Table 11: Default hyperparameter settings of attack methods.

| Methods | Hyperparameter | Settings |
|---|---|---|
| DC | Perturbation
Pre-trained model | $\ell_\infty = 8/255$
Official pretrained |
| NTGA | Perturbation
Poisoned dataset | $\ell_\infty = 8/255$
Official pretrained CIFAR-10 CNN (best) |
| EM | Perturbation
Perturbation type
Stopping error rate
Learning rate
Batch size
Optimizer | $\ell_\infty = 8/255$
Sample-wise
0.01
0.1
128
SGD |
| HYPO | Perturbation
Step size | $\ell_\infty = 8/255$
$\ell_\infty = 0.8/255$ |
| TAP | Perturbation | $\ell_\infty = 8/255$ |
| REM | Perturbation
Perturbation type
Stopping error rate
Learning rate
Batch size
Optimizer
Adversarial training perturbation | $\ell_\infty = 8/255$
Sample-wise
0.01
0.1
128
SGD
$\ell_\infty = 4/255$ |
| LSP | Perturbation
Patch size | $\ell_2 = 1.30$ (Project from $\ell_\infty = 6/255$)
8 for CIFAR-10/100 and SVHN; 32 for ImageNet |
| AR | Perturbation
Default hyperparameters | $\ell_2 = 1.00$
Follows official code |
| OPS | Perturbation
Perturbation type
Default hyperparameters | $\ell_0 = 1$
Sample-wise
Follows official code |
| UCL | Perturbation
Poisoned dataset | $\ell_\infty = 8/255$
Official pretrained CP-S of UCL |
| TUE | Perturbation
Poisoned dataset | $\ell_\infty = 8/255$
Official pretrained |

## C.3 VISUALIZATIONS

**Grad-CAM and Shapley visualizations** Gradient-weighted class activation mapping (Grad-CAM) [39] and Shapley value map [24] are commonly used image analysis tools that visualize the contributions of different pixels in an image to the model's predictions. From the Shapley value map and Grad-CAM visualizations (Figure 6), we can observe discernible changes in activation features in the poisoned model relative to the clean model. AVATAR showed activation features most similar to the clean model compared to other defense mechanisms, as it aims to restore the original features while disrupting the availability poisoning perturbations. Contrarily, models applying the other defense strategies typically have different activation features than clean models. This discrepancy implies that image preprocessing or augmentations may modify the inherent feature extraction from the original images instead of restoring them.

**T-SNE visualizations** Figure 8 shows the t-SNE visualization Figure 7 of the models' feature representations on the clean test set for CIFAR-10. Notably, models without defenses struggle to create coherent class clusters, although there exist spatial variations in class frequency. Conversely, models equipped with defenses display a feature distribution akin to the clean baseline. Models with higher clean accuracies often exhibit better separated clusters.

## C.4 ADDITIONAL RESULTS

Finally, Table 15 shows the detailed test accuracies of models trained on poisoned CIFAR-10 datasets, including an error range of 3 independent runs for each experiment.

Table 12: Default training hyperparameter settings.

| Datasets | Hyperparameter | Settings |
|---|---|---|
| CIFAR-10/-100 | Optimizer | SGD |
| | Momentum | 0.9 |
| | Weight-decay | 0.0005 |
| | Batch size | 128 |
| | Standard Augmentations | Random crop, random horizontal flip |
| | Training epochs | 50 |
| | Initial learning rate | 0.1 |
| | Learning rate schedule | Epochs per decay: 100, decay factor: 0.5 |
| SVHN | Optimizer | SGD |
| | Momentum | 0.9 |
| | Weight-decay | 0.0005 |
| | Batch size | 128 |
| | Standard augmentations | None |
| | Training epochs | 40 |
| | Initial learning rate | 0.1 |
| | Learning rate schedule | Epochs per decay: 100, decay factor: 0.5 |
| ImageNet-100 | Optimizer | SGD |
| | Momentum | 0.9 |
| | Weight-decay | 0.0005 |
| | Batch size | 256 |
| | Standard augmentations | Random crop, horizontal flip, and color jitter |
| | Training epochs | 100 |
| | Initial learning rate | 0.1 |
| | Learning rate schedule | Epochs per decay: 100, decay factor: 0.5 |

Table 13: Default hyperparameter settings of defenses.

| Methods | Hyperparameter | Settings |
|---|---|---|
| Adversarial training [25] | Perturbation | $\ell_\infty = 8/255$ |
| | Steps size | $\ell_\infty = 2/255$ |
| | PGD steps | 10 |
| | Training epochs | 200 |
| CutOut [6] / CutMix [46] / MixUp [47] | Training epochs | 200 |
| Gaussian [23] | Kernel size | 3 |
| | Standard deviation | 0.1 |
| | Training epochs | 200 |
| JPEG [23] | Quality | 10 |
| | Training epochs | 200 |
| BDR [23] | Number of bits | 2 |
| | Training epochs | 200 |
| UEraser-Lite [30] | PlasmaBrightness / PlasmaContrast | p = 0.5 |
| | ChannelShuffle | p = 0.5 |
| | Training epochs | 200 |
| UEraser-Max [30] | PlasmaBrightness / PlasmaContrast | p = 0.5 |
| | ChannelShuffle | p = 0.5 |
| | Number of Repeats $K$ | 5 |
| | Training epochs | 300 |
| AVATAR [7] | Diffusion sampler | Score-SDE |
| | Starting step / Total diffusion steps | 60 / 1000 |
| | Pre-trained model | Official pretrained |
| | Training epochs | 200 |

# D LIMITATIONS

APBench has mainly focused on providing algorithms and evaluations related to image data. However, such availability poisoning methods may also be applicable to text, speech, or video domains. In the future, we plan to expand APBench to include more domains, aiming to establish a more comprehensive and valuable benchmark for personal privacy protection against deep learning.

Table 14: Test accuracies (%) on CIFAR-10 PGD-20 adversarial examples.

| Type | Method | Standard | Gray | JPEG | U-Max | AVATAR | AT |
|------|--------|----------|------|------|-------|--------|-----|
| Black-box | EM | 16.83 | 19.77 | 73.85 | 71.43 | 20.07 | 69.77 |
| | REM | 22.53 | 19.16 | 73.70 | 80.46 | 17.36 | 70.74 |
| | LSP | 14.29 | 24.34 | 67.01 | 80.40 | 15.42 | 71.39 |
| | AR | 15.82 | 16.55 | 75.21 | 76.97 | 16.84 | 68.81 |
| White-box | EM | 0.00 | 0.00 | 3.74 | 1.37 | 0.00 | 28.60 |
| | REM | 0.00 | 0.00 | 3.35 | 3.11 | 0.00 | 32.52 |
| | LSP | 0.00 | 0.00 | 0.39 | 3.83 | 0.00 | 34.45 |
| | AR | 0.00 | 0.00 | 2.59 | 2.83 | 0.00 | 32.04 |

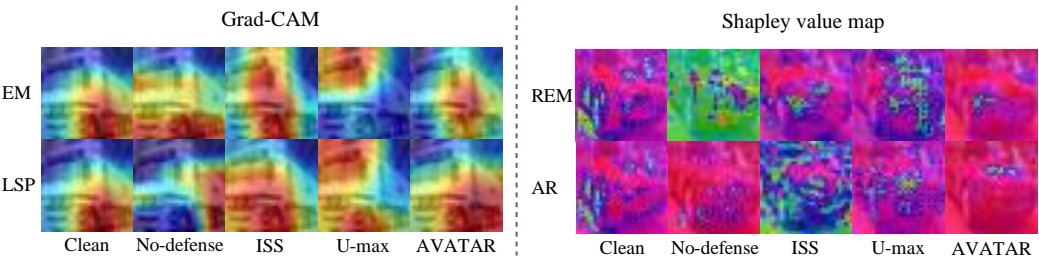

Figure 6: Grad-CAM and Shapley value map visualizations of regions contributed to model decision under different attack methods and defense methods with ResNet-18. (Left) Grad-CAM visualizations of EM and LSP attacks. (Right) Shapley value map visualizations of REM and AR attacks.

## E    ATTACK AND DEFENSE BASELINES

APBench is open source, and the source code will be made available upon publication. Table 16 provides the licenses of the derived implementations of the original algorithms and datasets.

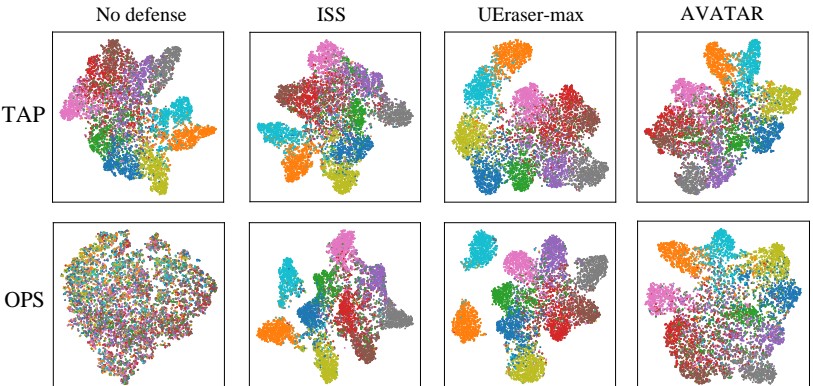

Figure 7: The t-SNE visualization of the models' feature representations on the clean test set. Note that without defenses, the feature representations of the poisoned models are mostly scrambled as the models struggle to learn useful features.

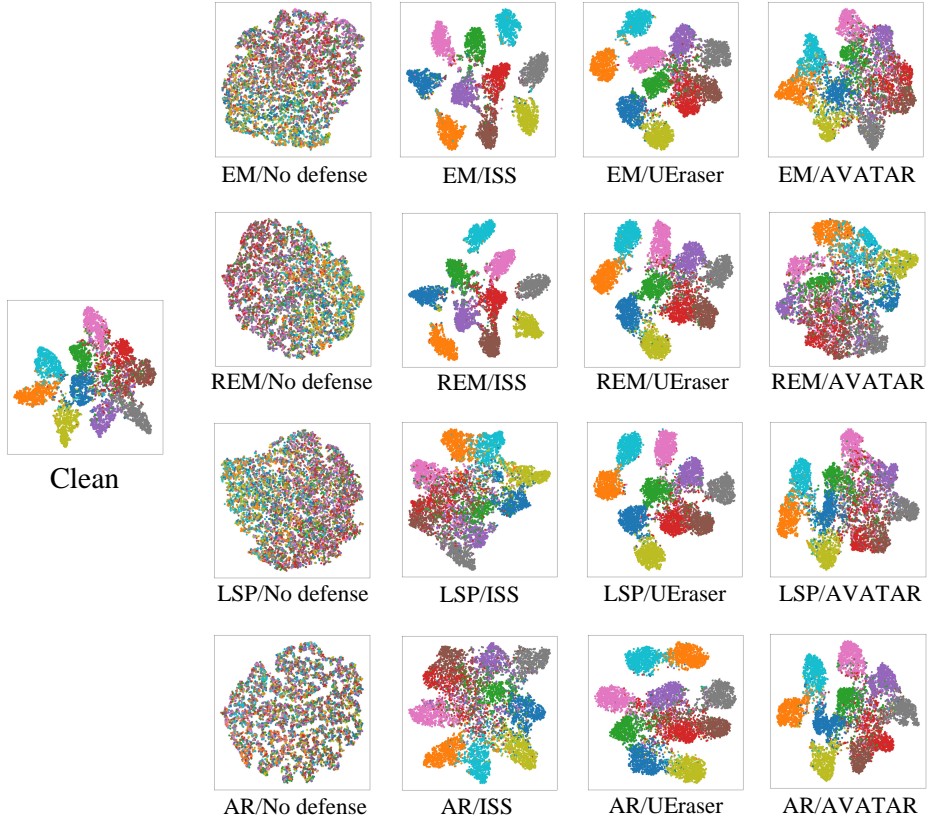

Figure 8: The t-SNE visualization of the models' feature representations on the clean test set under additional attacks for CIFAR-10. "UEraser" denotes "UEraser-Max".

Table 15: Detailed test accuracies (%) of models trained on poisoned CIFAR-10 datasets. This table is an extension of Table 3, and further includes an error range of 3 separate runs for each experiment.

| Method | Standard | CutOut | CutMix | MixUp | Gaussian | BDR | Gray | JPEG | U-Max | AVATAR | AT |
|---|---|---|---|---|---|---|---|---|---|---|---|
| DC | 15.19±1.87 | 19.94±1.90 | 17.91±2.66 | 25.07±2.74 | 16.10±1.07 | 67.73±2.77 | 85.55±2.04 | 83.57±2.61 | 92.17±0.97 | 82.10±2.20 | 76.85±1.79 |
| EM | 20.78±1.03 | 18.79±2.66 | 22.28±2.47 | 31.14±3.51 | 14.71±0.66 | 37.94±2.64 | 92.03±0.37 | 80.72±1.22 | 93.61±0.48 | 75.62±2.16 | 82.51±1.24 |
| REM | 17.47±2.04 | 21.96±3.72 | 26.22±2.86 | 43.07±4.36 | 21.80±0.94 | 58.60±2.35 | 92.27±0.29 | 85.44±2.10 | 92.43±0.61 | 82.42±1.47 | 77.46±1.66 |
| HYPO | 70.38±1.79 | 69.04±1.33 | 67.12±3.27 | 74.25±2.60 | 62.17±1.03 | 74.82±2.18 | 63.35±0.88 | 85.21±1.35 | 88.44±0.71 | 85.94±2.06 | 81.49±2.37 |
| NTGA | 22.76±0.67 | 13.78±0.75 | 12.91±1.22 | 20.59±2.41 | 19.95±1.26 | 59.32±1.96 | 70.41±2.67 | 68.72±3.37 | 86.78±1.64 | 86.22±2.07 | 69.70±2.66 |
| TAP | 6.27±0.48 | 9.88±0.71 | 14.21±2.14 | 15.46±1.77 | 7.88±0.66 | 70.75±1.46 | 11.01±0.67 | 84.08±2.36 | 79.05±1.04 | 87.75±2.52 | 79.92±1.87 |
| LSP | 13.06±0.74 | 14.96±1.02 | 17.69±1.37 | 18.77±2.12 | 18.61±0.82 | 53.86±2.40 | 64.70±3.29 | 80.14±2.51 | 92.83±1.27 | 76.90±1.09 | 81.38±1.92 |
| AR | 11.74±0.37 | 10.95±0.70 | 12.60±1.02 | 14.15±1.28 | 13.83±1.70 | 36.14±2.04 | 35.17±1.77 | 84.75±2.27 | 90.12±1.89 | 88.60±2.33 | 81.15±1.94 |
| OPS | 14.69±0.55 | 52.98±2.49 | 64.72±2.70 | 49.27±2.66 | 13.38±0.31 | 37.32±1.87 | 19.88±0.79 | 78.48±0.94 | 77.99±1.30 | 66.16±2.13 | 14.95±0.67 |

Table 16: Licenses of the datasets and codebases used in this paper.

| Name | License | URL |
|---|---|---|
| PyTorch | BSD | GitHub: pytorch/pytorch |
| DC | — | GitHub: kingfengji/DeepConfuse |
| NTGA | Apache-2.0 | GitHub: lionelmessi6410/ntga |
| EM | MIT | GitHub: HanxunH/Unlearnable-Examples |
| HYPO | MIT | GitHub: TLMichael/Delusive-Adversary |
| TAP | MIT | GitHub: lhfowl/adversarial_poisons |
| REM | MIT | GitHub: fshp971/robust-unlearnable-examples |
| LSP | — | GitHub: dayu11/Availability-Attacks-Create-Shortcuts |
| AR | MIT | GitHub: psandovalsegura/autoregressive-poisoning |
| OPS | Apache-2.0 | GitHub: cychomatica/One-Pixel-Shotcut |
| UCL | MIT | GitHub: kaiwenzha/contrastive-poisoning |
| TUE | — | GitHub: renjie3/TUE |
| ISS | — | GitHub: liuzrcc/ImageShortcutSqueezing |
| DiffPure | NVIDIA | GitHub: NVlabs/DiffPure |
| CIFAR-10 | — | https://www.cs.toronto.edu/~kriz/cifar.html |
| CIFAR-100 | — | https://www.cs.toronto.edu/~kriz/cifar.html |
| SVHN | — | http://ufldl.stanford.edu/housenumbers |
| ImageNet-100 | — | GitHub: TerryLoveMl/ImageNet-100-datasets |

