# OpenReview forum: "APBench: A Unified Benchmark for Availability Poisoning Attacks and Defenses"
_ICLR.cc/2024/Conference — Submitted to ICLR 2024_

### Official Review · Reviewer_gtEa · 2023-10-22

**Soundness:** 2 fair
**Presentation:** 1 poor
**Contribution:** 2 fair
**Rating:** 3
**Confidence:** 4

**Summary:**

This work proposes a benchmark of data poisoning attacks. The authors suggest an unified codebase where to test poisoning attacks and defenses.

**Strengths:**

+ Systematization effort for data poisoning attack evaluations

**Weaknesses:**

- the paper should be restructured
- does not compare with existing benchmarks
- limited contribution

**Questions:**

# Comments:

**The paper should be restructured**.
The sections of the paper are not disponed in a natural order. The authors should first describe the framework for systematizing data poisoning and defenses, then introduce the related work by matching the framework with the existing methods. This would increase the clarity of the paper, that otherwise is confusing and seems disconnected in its sections. The related work should be connected to the framework, otherwise rather than a benchmark this would simply be a collection of re-implemented methods.

**Does not compare with existing benchmarks / limited contribution.**
There are existing benchmarks and surveys on data poisoning, that the authors don't cite and compare with. The systematization itself is already done in other existing surveys that are not cited by this work. A non-exhaustive list is given below:

* Survey on data poisoning
  * https://arxiv.org/abs/2205.01992

* Other benchmarks on data poisoning that are not discussed
  * http://proceedings.mlr.press/v139/schwarzschild21a/schwarzschild21a.pdf
  * https://openreview.net/forum?id=PP3H72O_E2f
  * https://arxiv.org/abs/2009.02276 / https://github.com/JonasGeiping/poisoning-gradient-matching (implements also other poisoning methods)
  * https://github.com/JonasGeiping/data-poisoning

The authors should clarify what they add to the existing benchmarks, otherwise, highlighting the contributions that were not available to the community before this work. This also includes the fact that some of the techniques included in the benchmark were already available as source code, so the authors should clarify the implementation effort added to the existing tools.

Additional crucial weaknesses that should be addressed:
- limitations are not discussed
- the paper exceeds the page limit

---

> ### Author Response · Authors · 2023-11-21
> **Response to Reviewer gtEa**
>
> Thank you for reviewing our paper and we would like to address your concerns below.
>
> > The authors should first describe the framework for systematizing data poisoning and defenses, then introduce the related work by matching the framework with the existing methods. This would increase the clarity of the paper, that otherwise is confusing and seems disconnected in its sections.
>
> We appreciate the reviewer's suggestion.  We have updated the paper with revisions highlighted in blue to improve the clarity of the paper.  Regarding the organization of the paper, we incorporated the problem definition in Section 3, which formally defines the APA attackers and defenders, into Sections 2.1 and 2.2 respectively.  We believe this improves the flow of the paper and aligns with the reviewer's suggestion.  We also welcome any further suggestions to improve the clarity of the paper.
>
> > "This work proposes a benchmark of data poisoning attacks. The authors suggest an unified codebase where to test poisoning attacks and defenses." "Does not compare with existing benchmarks / limited contribution."
>
> Thank you for referencing the related works, and we have expanded the related data poisoning benchmarks and frameworks in Section 2.3.
>
> We believe there could be a misunderstanding regarding the purpose of availability poisoning attacks and defenses.
>
> We kindly point out that while APAs can be categorized as data poisoning attacks, there is a notable distinction between APAs and traditional data poisoning attacks regarding the threat model and the objectives.  The purpose of APAs is to protect the privacy of the attacker's public data against unauthorized training.  The goal of APAs is thus to make it difficult for deep learning algorithms to learn effectively from the data protected by APAs.  The intent of APAs is therefore benign rather than malicious as generally assumed of data poisoning attacks. For this reason, while traditional data poisoning aims to preserve clean accuracies and inject backdoors, models trained with availability poisons are expected to have low clean accuracies in order to achieve the goal of data privacy protection.
>
> In Section 2, we have expanded the problem definition (previously in Section 3) to explain the problem more clearly, and further clarify the differences between the two.
>
> As the referenced survey and benchmarks are for traditional data poisoning attacks, which are malicious in nature and aims to achieve backdoors while preserving clean accuracy, we believe they unfortunately cannot be directly implemented as APAs, and compared with existing APAs.
>
> > Additional weaknesses: limitations are not discussed, and the paper exceeds the page limit
>
> We kindly point out that Appendix D discusses the limitations of APBench and also per [ICLR 2024 Author Guide](https://iclr.cc/Conferences/2024/AuthorGuide), "This optional ethic statement/reproducibility statement will not count toward the page limit, but should not be more than 1 page".

---

### Official Review · Reviewer_WVAC · 2023-10-22

**Soundness:** 3 good
**Presentation:** 2 fair
**Contribution:** 2 fair
**Rating:** 5
**Confidence:** 4

**Summary:**

The paper studies availability poisoning attacks which add imperceptible perturbations to some samples to compute poisoning data and add it to the training data with the goal that the resulting model will perform poorly on any of the test data. In particular, the paper proposes a benchmark to evaluate various availability poisoning attacks and defenses; the benchmark is motivated from the fact that there are numerous attacks and defenses in the literature but no systematic framework or analysis of their comparisons. APBench has functionality to evaluate 9 attacks, 8 defenses, 4 data augmentations, and to ablate over different parameters of the attacks/defenses. Paper then provides evaluations of these attacks and defenses for 4 datasets and 4 model architectures, and finally an ablation study.

**Strengths:**

- Benchmarking is an important/useful step toward systemizing the knowledge in this area
- APBench seems to have comprehensive suite of attacks/defenses
- Paper has done thorough evaluation using APBench

**Weaknesses:**

- Threat model is difficult to understand
- Plenty of evaluations but insights are missing
- Clarity of writing can improve

**Questions:**

I think the benchmarking direction of the paper is very important and useful to researchers from academia/industry. I am not an expert in this area but I believe that the paper is very comprehensive in terms of incorporating multiple attacks/defenses. Finally, I think the paper has also done a great job of thorough evaluation. Given this, below are my concerns that I feel authors should address:

- I am not sure what is the threat model being studied. I noted that the paper mentions availability poisoning in different places but it would be good to have concrete details of the threat model, i.e., adversary’s goal, knowledge and capabilities.
    - Why such concrete threat model helps: In some parts of the paper APA is viewed as an attack that reduces model accuracy so it is a bad thing, but in some other parts, APA is (probably) viewed as a good thing because it protects private data from being learned by the model. My query might be naive but paper does not clarify it. Alternatively there might be two different settings where APAs are relevant: one where it is an attack that reduces accuracy and other where it is a defense that protects privacy. Authors should clarify these.
    - I also think the definition of AP in abstract is confusing: it says that AP is “a method of poisoning data by injecting imperceptible perturbations to prevent its use in model training”; my question: why do you even add such points to data? Maybe you want to say that you add these perturbed data to reduce model’s overall performance; which is what Eq (1) implies. But then the end of the abstract says “Our results reveal the glaring inadequacy of existing attacks in safeguarding individual privacy”, which sounds more like these attacks are deployed for something good, i.e., protecting individual privacy.
    - Eq (1) implies that AP is basically an untargeted attack that just aims to reduce model’s accuracy; if so, maybe explain this somewhere (preferably in threat model) for ease of understanding.
    - Overall I think the motivation of the attacks in this paper needs proper justification and there should a place in the paper for threat models/settings considered in the paper.
- I noted that there is a lot of evaluation in the paper, but I don’t know what insights to draw from them.
    - Evaluations feel like long blocks of text. I suggest highlighting useful text to help readers understand what in the final conclusion to draw from an eval.
    - At least some of the evaluations should highlight how APBench will help a practitioner identify something that prior works alone cannot identify. For example, in “Larger perturbations”, the conclusion is that “There exists a trade-off between perturbation magnitude and accuracy recovery”. This feels like an obvious conclusion; the paper should highlight what is it that APBench brings to the light that prior works could not.
- Paper is a bit difficult to read; I feel all the pieces are there in the paper but are not correctly placed/organized/explained:
    - In “privacy protection under partial poisoning”:
        - “As can be seen in Figure 3… the whole dataset”: this seems wrong; how can accuracy increase with more poisoning?
        - Mean losses of Figure 4: are these over training steps/ samples?
        - Can you rephrase the question: “can the protective perturbation… model training?” This question probably is about the good use of APAs; please clarify that as well.
        - “We find that the losses on the original… private data against learning”: If the private data is not supposed to be learned, why is it in the training data in the first place? Please clarify.
    - In “Future outlook”:
        - Future methods should enhance the resilience of perturbation: I am not sure if this is for good (improve privacy) or bad (improve attacks)? Clarify.

---

> ### Author Response · Authors · 2023-11-21
> **Response to Reviewer WVAC**
>
> Thank you for reviewing our paper and we would like to address your concerns below.
>
> > Clarifications on the threat model.
>
> Since APAs can be categorized as data poisoning attacks, we follow existing naming conventions which refer to the "attacker" as the one that adds the perturbations, and the "defender" as the one that trains the model.  In that sense, "poisoning" actually refers to the attacker adding perturbations to the data to hinder the defender's ability to train models.
>
> With the above definitions, we can summarize the threat model as follows: We assume that the attacker has a set of data which they wish to publish while protecting them against machine learning.  By introducing small perturbations into the data, the attacker aims to hinder the trainer's (defender's) ability to train models that can generalize effectively to the original data distribution from the protected data.
>
> While both traditional data poisoning attacks and APAs add small perturbations to the data, the intention of the APAs is *not* to introduce stealth backdoor behaviors in models while preserving clean accuracies, but rather to reduce the ability of the models to learn from the data with crafted perturbations, in order to protect the attacker's privacy.
>
> We have also since expanded Section 3 to explain the threat model in greater detail, and further clarify the differences between the related benchmarks in Section 2.
>
> > I suggest highlighting useful text to help readers understand what in the final conclusion to draw from an eval.
>
> > At least some of the evaluations should highlight how APBench will help a practitioner that prior works alone cannot identify.
>
> Thank you for your constructive suggestions!  We have highlighted the valuable insights with italics in the updated version.  We also kindly point out that the main key takeaway of APBench is it is the first benchmark to shed light on the inadequacy of existing APAs against stronger defenses.
>
> > "As can be seen in Figure 3 ... the whole dataset."
>
> Thank you for noticing this!  We meant to claim that "the test accuracy of the model in the case of partial poisoning is only slightly lower than that in the case of *a completely clean dataset*," instead of "*poisoning the whole dataset*." The paper has been updated to reflect this.
>
> > Mean losses of Figure 4: are these over training steps/ samples?
>
> For the mean losses, we take the final model after training, and evaluate the mean loss values by averaging the loss values of the model for each image in the respective datasets.
>
> > Can you rephrase the question: "can the protective perturbation… model training?"
>
> We agree with the reviewer the question can be rephrase be more readable.  For this we have updated the question to be as follows in the paper: "*Are APAs effective in protecting only a portion of the training data?* To answer, we introduce poisoning perturbations with APAs to a varying portion of the training data, and investigate how well the models learn the origin features that exist in the poisoned images for different poisoning rates."
>
> > If the private data is not supposed to be learned, why is it in the training data in the first place?
>
> Quoting from [1], "Personal data has been unconsciously collected from the Internet and used to train commercial models, raising public concerns about the 'free' exploration of personal data for unauthorized or even illegal purposes." As none of us are immune to having our private data collected, perturbations can be added with APAs in order to protect data against unauthorized training.
>
> [1] Huang, Hanxun, et al. Unlearnable examples: Making personal data unexploitable, 2021
>
> > Future methods should enhance the resilience of perturbation: I am not sure if this is for good (improve privacy) or bad (improve attacks)? Clarify.
>
> We hope it is clear by now that APAs are meant to protect data against unauthorized training.  Defenses explored in this paper are used to evaluate the effectiveness of APAs.

---

### Official Review · Reviewer_nmU1 · 2023-10-30

**Soundness:** 1 poor
**Presentation:** 3 good
**Contribution:** 2 fair
**Rating:** 3
**Confidence:** 5

**Summary:**

This paper acknowledges the challenges in evaluating availability poisoning attacks and introduces an open-source benchmark, APBench, which comprises various poisoning attacks, defense strategies, and data augmentation techniques. The benchmark is designed to facilitate fair and reproducible evaluations, revealing shortcomings in existing attacks and promoting the development of more robust defense methods. Ultimately, APBench serves as a platform to advance availability poisoning attack and defense techniques, aiming to protect privacy and data utility.

**Strengths:**

- Diverse experiments
- Visual analysis results
- Encourages future research

**Weaknesses:**

- Inadequate explanation for attack selection
- Setup lacks explanation
- Inconsistent results
- Conclusions that appear to be at odds with the available evidence
- Incomplete defense evaluation

**Questions:**

- **Inadequate explanation for attack selection.** The authors' rationale for selecting specific attacks in the context of availability poisoning requires further clarification. Since the aim is to establish a benchmark for general availability poisoning attacks, it is advisable to include a broader spectrum of attack types, such as those referenced in citations [1], [2], [3], and [4]. Currently, the focus appears to be primarily on clean label poisoning attacks, which may not align with the assumptions of availability poisoning attacks, which do not assume clean labels. Additionally, the presented attacks predominantly target deep learning networks, the authors should either tune down their scope or include attacks against other learning algorithms like SVM.

- **Setup lacks explanation.** In Section 5.1, during the evaluation of partial poisoning, it would be beneficial for the paper to provide clarity regarding the methodology employed for generating these partial poisons. Specifically, it would be valuable to know whether the process involves the initial generation of poisons using all clean data, followed by the selection of a subset from these generated poisons. Or do we solely utilize a portion of the data from the beginning to the end? It is important since training the surrogate model also necessitates clean data.

- **Inconsistent results.** In Figure 3 and 4, it is evident that both Greyscale and JPEG consistently exhibit a similar impact, demonstrating uniform performance across all attack methods and poisoning rates. However, this consistency appears to contrast with the performance discrepancies observed in other tables within the paper. It would be valuable if the authors could offer an explanation for this observed consistency

- **Conclusions that appear to be at odds with the available evidence.** In Section 5.1, the authors claim that in partial poisoning scenarios, test accuracy only slightly decreases compared to poisoning the entire dataset. However, it's important to support this claim with experimental evidence. Specifically, I reference the original paper on EM, which reveals that when 80% of the data is poisoned, the model maintains a test accuracy slightly above 80%, but this accuracy significantly drops to less than 20% when the entire dataset is subjected to poisoning. This substantial disparity in accuracy between partial and full dataset poisoning underscores the need for the authors to revise their statement and include a poison rate of 100% as a baseline in Figure 3 for a more comprehensive analysis.

- **Incomplete defense evaluation.** While the authors mention considering various existing defenses, the primary focus is on data preprocessing methods. It is advisable to provide more extensive results for training-phase defenses, which have proven effective in defending data poisoning. For instance, common defenses like adversarial training are briefly mentioned but not thoroughly discussed. Also, early stopping has shown its effectiveness in many cases. These defense mechanisms have been explored in previous work like [5] and [6] and should be given more attention. Alternatively, the scope of the paper could be refined to focus specifically on data preprocessing defenses to maintain coherence.

[1] Poisoning Attacks against Support Vector Machines

[2] Towards Poisoning of Deep Learning Algorithms with Back-gradient Optimization

[3] Preventing Unauthorized Use of Proprietary Data: Poisoning for Secure Dataset Release

[4] Witches' Brew: Industrial Scale Data Poisoning via Gradient Matching

[5] Is Adversarial Training Really a Silver Bullet for Mitigating Data Poisoning

[6] Poisons that Are Learned Faster are More Effective

---

> ### Author Response · Authors · 2023-11-21
> **Response to reviewer nmU1**
>
> Thank you for reviewing our paper and we would like to address your concerns below.
>
> > Inadequate explanation for attack selection.
>
> We kindly point out that while APAs can be categorized as data poisoning attacks, there is a notable distinction between APAs and traditional data poisoning attacks ([1], [2], [4]) regarding the threat model and the objectives.  The goal of APAs is to make it difficult for deep learning algorithms to learn effectively from the data protected by APAs. The intent of APAs is therefore benign rather than malicious as generally assumed of data poisoning attacks.
>
> In contrast, traditional data poisoning injects triggers into a small amount of data in a dataset, causing any input with the injected trigger to be misclassified by a model trained on the poisoned dataset.
>
> Perhaps the reviewer could explain why "the assumptions of availability poisoning attacks ... do not assume clean labels".
> In general, existing APAs assume the training party has the ability to curate the correct labels for all images.
>
> In Section 2, we have expanded the problem definition (previously in Section 3) to explain the problem more clearly, and further clarify the differences between the two.
>
> > Setup lacks explanation.
>
> Thank you for your raising this question.  We consider attackers who can initially generate poisons using all clean data and then select a subset of these generated poisons.  For APA attacks that require a surrogate model, we assume that the attackers have access to the full training data, to make the attacks stronger and to also compare them fairly.  As for the surrogate-free APA attacks (AR and LSP), no training data is required as they generate strongly linear separable perturbations algorithmically.
>
> > Inconsistent results.
>
> Thank you for noticing this!  We found that Figure 3 mistakenly repeats the Grayscale results for JPEG defenses, and have fixed this mistake in the updated version.
>
> > Conclusions that appear to be at odds with the available evidence.
>
> Again, thank you for noticing this!  We fully agree with the reviewer, and we meant to claim that "the test accuracy of the model in the case of partial poisoning is only slightly lower than that in the case of *a completely clean dataset*," instead of "*poisoning the whole dataset*."
>
> > These defense mechanisms have been explored in previous work like [5] and [6] and should be given more attention.
>
> Thank you for your kind suggestion.  [5] explores a similar setting as the HYPO attack examined in this paper.
>
> We have now included early stopping (ES) as considered in [6] in the updated paper as a training-phase defense, and we report the peak accuracies achieved in Tables 3, 4, 5 and 6.  We found that as a defense it is not very effective against most APA attacks in comparison to other defenses.

---

### Official Review · Reviewer_8Vbt · 2023-10-31

**Soundness:** 3 good
**Presentation:** 3 good
**Contribution:** 3 good
**Rating:** 6
**Confidence:** 4

**Summary:**

This paper presents APBench, a unified benchmark designed to evaluate the efficacy of availability poisoning attacks and defenses in machine learning. Availability poisoning attacks subtly manipulate training data to disrupt model learning, created to mitigate concerns about data security and privacy. APBench addresses this by offering a standardized platform, including 9 advanced poisoning attacks, 8 defense algorithms, and 4 data augmentation techniques, facilitating comprehensive evaluations across different datasets, model architectures, and poisoning ratios. The results highlight the limitations of existing attacks in ensuring privacy, underscoring the need for more robust defenses. APBench is open-sourced, aiming to foster advancements in secure and privacy-preserving machine learning.

**Strengths:**

- Comprehensive Benchmarking. APBench provides a comprehensive and unified benchmark for evaluating availability poisoning attacks and defenses, addressing a crucial gap in the field. It includes a wide array of state-of-the-art poisoning attacks, defense algorithms, and data augmentation techniques, ensuring its relevance and applicability to a broad spectrum of scenarios.

- Extensive Evaluation. The paper conducts extensive evaluations across various datasets, model architectures, and poisoning ratios. This thorough testing ensures that the results are robust and reliable, providing valuable insights into the effectiveness of different attack-defense combinations.

**Weaknesses:**

- While APBench provides a comprehensive evaluation of availability poisoning attacks and defenses, it is limited to this specific type of adversarial attack. Other types of adversarial attacks, such as integrity or backdoor attacks, are not covered. This direction would be interesting. For example, the defenses that have been developed for resisting backdoor attacks might be capable of mitigating the existing availability attacks. This phenomenon is possible, since it has been observed that the best defense against test-time adversarial examples is also a principled defense against availability poisoning attacks. A more unified benchmark can facilitate the discovery of these possibilities and opportunities.

- The TAP attack should be partially attributed to Nakkiran [1], who was the first to point out such class targeted attacks. The unique contribution of [2] is that they found that untargeted adversarial examples can also be an effective availability poisoning attack. (This information is recorded on page 4 of the camera-ready version of [2].)

[1] Nakkiran, "A Discussion of 'Adversarial Examples Are Not Bugs, They Are Features': Adversarial Examples are Just Bugs, Too", Distill, 2019.
[2] Fowl et al., Adversarial examples make strong poisons, NeurIPS 2021.

**Questions:**

- How might defenses originally developed for backdoor attacks, e.g. ANP [3], be effectively repurposed to mitigate availability attacks?

- This benchmark exclusively focuses on evaluating the natural test accuracy of the trained models. How about their test robustness, e.g., PGD-20 accuracy?


[3] Wu & Wang, Adversarial Neuron Pruning Puriﬁes Backdoored Deep Models, NeurIPS 2021.

---

> ### Author Response · Authors · 2023-11-21
> **Response to Reviewer 8Vbt**
>
> Thank you for reviewing our paper and acknowledging its contribution.  We would like to address your concerns below.
>
> > defenses that have been developed for resisting backdoor attacks might be capable of mitigating the existing availability attacks
>
> > How might defenses originally developed for backdoor attacks, e.g. ANP [3], be effectively repurposed to mitigate availability attacks?
>
> Thanks for your suggestion!  We agree with the reviewer that backdoor attacks may present a promising avenue to mitigate availability attacks.  However, due to differences in the objectives, most of the current backdoor defense methods require a certain number of clean samples (including ANP as kindly suggested by the reviewer), which may subjectively violate the assumed capabilities of the defender.
>
> > The TAP attack should be partially attributed to Nakkiran.
>
> Thanks for pointing this out, we have updated this description in the updated version.
>
> > How about their test robustness, e.g., PGD-20 accuracy?
>
> Thank you for the suggestion, and we have added this evaluation in Table 14 of Appendix C.2 of the updated version.

---

### Meta-Review · Area_Chair_StyM · 2023-12-12

**Metareview:**

This paper looks at benchmarking for poisoning attacks.  This is a very nuanced (and important) space in ML security.  Reviewers appreciated the motivation for the work in addition to the approach and the timeliness of the work.  Yet, across the board there were concerns of comprehensive coverage of the space of attacks in addition to metrics; these were partially but not fully addressed by the authors' rebuttal.  The unilateral belief was that the paper is a strong but yet relatively immature foundation for a great benchmark+paper in the coming submission cycles.

**Justification For Why Not Higher Score:**

Rebuttal did not address the majority of reviewer concerns.

**Justification For Why Not Lower Score:**

N/A

---

### Decision · Program_Chairs · 2024-01-16

Reject